# Opinion: Gigacity – a source of problems or the new way to sustainable development

Markku Kulmala[1,2,3], Tom V. Kokkonen[1,2], Juha Pekkanen[4,5], Sami Paatero[2], Tuukka Petäjä[1,2,3], Veli-Matti Kerminen[2], Aijun Ding[1]

[1]Joint International Research Laboratory of Atmospheric and Earth System Sciences, School of Atmospheric Sciences, Nanjing University, Nanjing, China
[2]Institute for Atmospheric and Earth System Research / Physics, Faculty of Science, University of Helsinki, Finland
[3]Aerosol and Haze Laboratory, Beijing Advanced Innovation Center for Soft Matter Science and Engineering, Beijing University of Chemical Technology, Beijing, China
[4] Department of Public Health, University of Helsinki, P.O.Box 20, 00014 University of Helsinki, Finland
[5] Environmental Health Unit, Finnish Institute for Health and Welfare, Helsinki, Finland

*Correspondence to*: Markku Kulmala (markku.kulmala@helsinki.fi) and Aijun Ding (dingaj@nju.edu.cn)

**Abstract.** The eastern part of China as a whole is practically a gigacity, a conglomeration of megacities with ca 650 000 000 inhabitants. The gigacity, with its emissions, processes in pollution cocktail, numerous feedbacks and interactions, has a crucial and big impact on regional air quality within itself as well as on global climate. A large-scale research and innovation program is needed to meet the interlinked grand challenges in this gigacity and to serve as a platform for finding pathways for sustainable development of the whole Globe.

## 1 Grand challenges and urban development

Humanity faces a multitude of severe global environmental changes, such as climate change, air, water and soil pollution, disturbances to food and water supplies and global epidemic diseases (Figueres et al., 2017; Burnett et al., 2018; Parrish et al., 2009; Zhang et al., 2017). On a global scale, these environmental challenges are called "Grand Challenges", which are the main factors controlling human well-being and security as well as the stability of future societies. Since Grand Challenges are highly connected and interlinked, they cannot be solved separately (Fig. 1). The main driving forces behind these are the growth of population and gross domestic production (GDP) globally, as well as the increasing urbanization, which is closely related to the former two.

Cities provide the citizens with many health services. These are, however, being counteracted e.g. with increasing risks of cardiovascular diseases and diabetes, violence and injuries, outbreaks of infectious disease, like the current pandemic of COVID-19 (Tian et al., 2020), and inequity between people living in urban areas. These problems are especially acute in large

and rapidly growing cities, which concurrently struggle to build sufficient infrastructures to provide clean air and water, energy, food, transportation, waste management and public spaces, all being essential to human well-being (Yang et al., 2018).

The urban sprawl is a major contributor to the destruction of natural, biologically diverse habitats and the current rapid decline in biodiversity (e.g. Liu et al. 2016). The cities rely on the surrounding country-side for several immediate ecosystem services (Ramaswami et al., 2016), and a limited contact with nature has been shown to reduce human wellbeing. There are some indications that the loss of biodiversity favors the emergence of new infectious diseases and antibiotic resistance. In the long run, the survival of mankind depends on a diverse and functioning ecosystem.

A clear example of future development is the rapid, large-scale urbanization of China, being so far unique in history (Zhao et al., 2015). Already now about 10% of the global population is living in the area of 1 Mkm$^2$ in eastern China inside the triangle confined roughly by the lines Shanghai-Nanjing-Xian-Beijing-Shanghai (Fig. 2). The population is still increasing and gradually filling the less urbanized areas between the individual cities, so that this region is becoming practically one city –

45 termed gigacity here – particularly from the grand challenges point of view: clean air is lacking, the frequency of severe weather events with human casualties is increasing, biodiversity is going down and sources of food and water are polluted (Fang et al., 2018; Ding et al., 2017; Kulmala, 2015; Lu et al., 2019), with all this happening concurrently with an increasing demand for water, food and energy. Since this gigacity has roughly 50 times more people and 60 times larger surface area than Beijing – a typical megacity – its future is crucial not only for local people but also globally. The area is a huge emitter of

50 greenhouse gases and air pollution as well as a potential source for local, regional and global epidemiological diseases due to loss of biodiversity and a huge population density. What will happen there in the future, will affect the whole Globe.

On a positive note, as a starting point of the new Silk Road and economic belt, the Chinese gigacity could serve as an example for other similar areas in the future on how to meet and even solve the grand challenges. On one hand, the Gross Domestic

Product (GDP) of China has increased at a rate of several percents per year during the past few decades. On the other hand, the Chinese Government has estimated that the GDP is reduced by 4–8 % due to air pollution, and it is very likely that other grand challenges have the similar impact, which underlines that there are economic incentives to solve the interconnected Grand Challenges.

## 2 Challenges specific for the gigacities

Air pollution is recognized as the biggest environmental challenge in China today (Huang et al., 2021; Parrish et al., 2009; Yang et al., 2018; Zhang et al., 2017) and it is estimated to cause more than a million deaths annually in China (Yang et al., 2018; Zhang et al., 2017). Fine particulate air pollution, as the most harmful air pollutant (Parrish et al., 2009; Zhang et al., 2017, Burnett et al., 2018) reduces life expectancy and increases the risk of cardiovascular and respiratory diseases. Transport

causes 20 to 40% of air pollutant emissions in eastern China (Huang et al., 2014: Huang et al., 2021). Therefore, reduction of the amount of motorized transport could improve air quality, but such reductions might even cause negative impact on the air quality due to non-balanced reduction, as was demonstrated e.g. by Huang et al. (2021) with the data obtained during the COVID-19 lockdown in China. It is already well known that reduced concentrations of nitrogen oxides will, under otherwise polluted conditions, increase ozone production and increase secondary aerosol particle concentrations (Ding et al., 2013, Liu and Tang, 2020). The air quality can be further deteriorated due to aerosol-boundary layer-weather feedback (Huang et al., 2020). In the gigacity domain, non-linear interactions between the atmospheric composition, meteorology, city planning/land use, human adaption and behavior need to be understood in a comprehensive and holistic way (Baklanov et al., 2018). Compared with megacities, the synergic effects of such interactions may result in much higher exposures and health effects in a gigacity. Future research is needed into this direction.

Although high-resolution modeling is capable of forecasting air quality and study the urban climate phenomena inside a megacity with a grid scale of about 10 meters, incorporating local-scale phenomena into a gigacity scale is not straight forward as it requires scaling from 100 km up to more than 1000 km (Huang et al., 2020; Huang et al., 2021). This calls for a seamless modeling approach (Baklanov et al., 2018) that couples the urban atmosphere and hydrology driven by synoptic-scale numerical weather prediction models, or even Earth System models. This further allows for bridging the gigacity environment to a continental scale, while retaining the micro- and meso-scale process-level understanding. Regional-scale climate and air quality are influenced by sources both outside and inside of the gigacity area, such as windblown dust, biogenic emissions, biomass burning, traffic and industry (Ding et al., 2017; Huang et al., 2020). On a larger scale, the gigacity is a huge source of atmospheric pollution and anthropogenic heat. A combination of Urban Heat Island (UHI), increased surface roughness and aerosol-Planetary Boundary layer interaction will influence not only the haze pollution within the urban areas but also precipitation patterns, hydrological cycle, and regional-scale weather patterns for example by disrupting and bifurcating storm fronts (Dou et al., 2015). However, gigacity acting as one huge urban area has a potential to disrupt even larger scale weather patterns, such as Asian monsoon, thereby influencing even global climate.

## 3 Feedback mechanisms in the gigacity

In a typical megacity suffering from atmospheric pollution, haze formation, together with Planetary Boundary Layer (PBL) processes, include a suite of feedback mechanisms and non-linear interactions. The haze reduces the amount of solar radiation reaching the surface (Arnfield, 2003), which decreases the near-surface air temperature and thereby reduces the vertical turbulent mixing of air and consequently daytime BL height (Ding et al., 2017; Ma et al., 2020; Petäjä et al., 2016; Wang et al., 2020). During the winter 2018 in Beijing, we found that compared with clean days, the reduced incoming solar radiation on the haze days decreased the BL height on average by 1660 m during daytime (Fig. 3a). The daytime BL height decreased by approximately 70% when the $PM_{2.5}$ concentration increased by a factor of 10 (Fig. 3b). At the same time, solar heat stored

into urban built-up surfaces was reduced due to the attenuated incoming solar radiation on haze days, which weakened the nocturnal UHI by about 1.6 K (Fig. 3a), while the temporal evolution of UHI during daytime remained relatively unchanged. The average magnitude of nighttime UHI decreased of about 40% when the $PM_{2.5}$ concentration increased by a factor of 10 (Fig. 3b). In addition, the reduced nighttime UHI and heat emissions produced lower nocturnal BL during haze days (on average by 490 m) (Fig. 3a). As a summary, in a typical megacity the feedbacks associated with the BL height and particulate

pollution will force the emissions into a smaller and smaller volume, further enhancing the accumulation of pollutants and the overall weakening of the UHI will decrease the urban heat dome flow, inhibiting the lateral ventilation of urban areas (Miao et al., 2015) on a regional scale.

In typical urban areas, the maximum values of UHI are restricted by the UHI induced circulation between the rural and urban areas, bringing cooler and cleaner air into the city. However, within the gigacity multiple smaller-scale UHI circulations are

formed, leading to almost closed internal circulation patterns relying on warm and polluted air (Fig. 4) as the influx. The UHI dynamics is highly dependent on the season, latitude, local climate, relative geographical location, and the size and shape of the city (Han et al., 2020; Huang et al., 2020). Therefore, the details remain unclear. However, we can foresee that an enhanced frequency and intensity of extreme rainfall and flash floods within and downwind of the gigacity and a decreased amount of total cumulative rainfall (Mahmood et al., 2014; Zhang et al., 2019) are likely to occur. These changes are connected to the

increased surface roughness as well as modified energy and water balances associated with the combination of UHI, high concentration of aerosols and higher fraction of constructed regions. Such consequences pose challenges to the air quality, water supply and agricultural production within the gigacity (Ding et al., 2017). Due to the non-linear nature of these interactions, the consequences can be stronger, or even reversed, in the gigacity compared with an isolated megacity.

The atmospheric pollution at the gigacity area is a complex mixture of local and regional-scale phenomena (e.g. Harrison et

al., 2021; Huang et al., 2020; Kulmala, 2015; Wu et al., 2020). On one hand we have emissions from the local emission hotspots within the gigacity (e.g. industrial sources), as well as local and regional-scale secondary aerosol formation (Huang et al., 2020). On the other hand, we have regional transport of pollutants within the gigacity and transport from outside the gigacity area (e.g. industry, biomass burning etc.) (Ding et al., 2017). Many recent studies (e.g. Huang et al., 2018) have demonstrated that the impact of anthropogenic processes, such as those related to air pollution, have not been included in the

nowadays weather forecast models. As a result, these models show notable biases in the prediction of free-tropospheric air temperature in the gigacity region. In fact, in the gigacity region, anthropogenic aerosols (e.g. black carbon) could significantly influence the development of PBL via not only the reduction of surface solar radiation but also its "dome effect" by heating the upper-PBL (Ding et al., 2016; Wang et al., 2018). In addition, aerosol-PBL feedback could also occur at the gigacity scale by amplifying the transboundary transport of haze, including secondary pollutants, between different megacity clusters, such

as the Yangtze River Delta and the Beijing-Tianjin-Hebei Area (Huang et al., 2020).

The Chinese government has started to tackle the air pollution with the Clean Air Act from 2013 onwards, which has already shown substantial reduction of pollutants in eastern China (Ding et al., 2019; Vu et al., 2019). However, non-balanced reductions in otherwise polluted regions could lead even to a negative effect as was demonstrated during the COVID-19 lockdown (Huang et al, 2021). Therefore, further research is needed in order to better understand the synergic effects of multi-pollutant emission reduction under the influence of regional-scale aerosol-PBL-weather interaction (Ding et al., 2017).

Even though the reduction of pollutants will lead also to increased BLH, which will reduce the pollution further, the cleaner air will allow also more incoming solar radiation to reach urban surfaces, which will increase the UHI effect in the gigacity area. Already now the individual megacities within the gigacity are suffering from extreme heat during the summertime (e.g. Nanjing). An increase of UHI and the lack of cooler air as an input flux from the diminishing rural areas could lead to unbearable urban summertime temperatures inside the gigacity. Therefore, a comprehensive research program would be needed in order to understand the interactions of the interlinked local and regional-scale phenomena, the associated feedbacks, and the highly complex atmospheric chemical processes.

## 4 Future research needs in the gigacity

We need an integrated research program comprising all the scales from understanding processes in atmospheric pollution cocktail via local boundary layer dynamics and regional air quality to even global scale in order to understand the effect of the whole gigacity region on the local/regional air pollution and larger-scale weather patterns and climate. For example, anthropogenic effects (urban structures, air pollution, anthropogenic heat) have been shown to modify substantially the water balance in a neighborhood scale (Best and Grimmond, 2016; Grimmond and Oke, 1986; Kokkonen et al., 2018a, 2019) and in the scale of the whole city (Dou et al., 2015). Urban areas can also disrupt and bifurcate storm fronts (Dou et al., 2015) and because the gigacity is a huge source of anthropogenic heat, air pollution and a vast area of increased surface roughness, it has a potential to affect Asian monsoon and therefore definitely regional but even global climate.

The Asian monsoon is very important for the whole gigacity region because disturbances in the Asian monsoon can affect the fresh water availability in the region and can also cause catastrophic flooding (Ding et al., 2015; Li et al., 2016). However, as the focus of the ongoing and past research has been on the effects of the global climate change on the Asian monsoon (Li et al., 2016; Zhang, 2016; Goodkin et al., 2019; Seth et al., 2019), we need to understand better how all these anthropogenic phenomena, e.g. the effects of urban structures, air pollution, anthropogenic heat and also anthropogenic climate change, affect together on the very large scale of the gigacity and its downwind areas.

Currently, detailed studies within the gigacity are typically covering the different phenomena, such as weather forecast, urbanization or air quality, as separate issues. As mentioned above, in gigacity areas human activities could influence the

climate system all the way from the nanoscale to the global scale. The lack of comprehensive studies is partly due to the lack of openly available comprehensive observation data. If suitable observations are lacking either high enough spatial coverage or some of the needed variables, one option is to use reanalysis data. However, such data are not available in high enough resolution and they do not describe urban areas, especially local-scale air pollution, in high enough accuracy for local-scale urban studies (Best and Grimmond, 2016; Fowler, et al., 2007; Kokkonen et al., 2018b, 2019).

Therefore, we need comprehensive observations and high-resolution fully-coupled models to accurately describe the molecular-scale chemistry, microphysics of aerosol-cloud-interaction, and the multi-scale processes of aerosol-PBL-weather feedback at the gigacity and even global scale. The spatial coverage of observations and the model resolution needs to be high enough to well characterize the complexity of emissions and chemistry, land-surface processes, and their interaction with the PBL meteorology. We also need to understand aerosol physics and atmospheric chemistry in molecular and nm scales, as the majority of particle number and mass concentrations, as well as ozone, is caused by secondary processes taking place in the atmosphere (e.g. Kulmala et al., 2021). To understand such secondary processes, we need proper observations and process level models – including quantum chemistry (e.g. Kulmala et al., 2021). Since some of the processes might be unique to the gigacity area, and possibly still unknown, we need comprehensive observations and concurrent models that cover different regions of the gigacity (Kulmala 2015, 2018), in order to characterize different processes and their interactions and to force the models and validate the results.

The different aspects of a comprehensive research and innovation program for the gigacity is presented in Fig. 5. The program should include all the potential emissions and explore pathways for their reductions. It should also include atmospheric processes, air composition and concentrations, atmospheric dynamics, as well as pollution levels outdoors and indoors. It is crucial also to include environmental, health, economy and societal impacts and decision making processes. The decision making will then feedback to the emissions and influence air quality indoors and outdoors.

The aims and main research questions of the program should include:
   a) To quantify emission patterns in high spatio-temporal resolution and to find pathways to reduce emissions that are most harmful for the pollution cocktail,
   b) To understand processes in the atmospheric pollution cocktail in order to reduce secondary pollution within the gigacity,
   c) To understand the effect of planetary boundary layer dynamics in enhancing haze episodes in a gigacity scale and furthermore to find out ways to avoid them,
   d) To understand the urban heat island (UHI) and its dynamics, and to reduce the adverse impacts of both UHI and pollution in the gigacity scale,
   e) To understand air pollution – weather interactions and feedbacks,

f) To understand and quantify air pollution – climate feedbacks and interactions,

g) To find out and quantify the contribution of the gigacity to the global climate,

h) To find out sustainable ways to solve the other grand challenges, such as sustainable water and food supply,

i) To explore biodiversity – air quality feedbacks,

j) To find out air quality – pandemic interactions and feedbacks,

k) To find out pathways to protect biodiversity within the gigacity,

l) To quantify non-linear interactions between the atmospheric composition, meteorology, city planning/land use, human adaption and behavior.

In addition, it is already foreseen that as soon as we have comprehensive open data sets available from the gigacity environment, we are able to pose questions that we not yet have identified and provide quantitative answers to the global grand challenges.

## 5 Other potential gigacity areas

At the moment there are no other truly gigacity areas in the world where the cluster of individual megacities could be classified as one huge continuous urban area. However, there are already other possible areas known that might struggle with similar problems in the future as the Chinese gigacity. Especially the Ganges basin with a population of about 400 million people is definitely already quite close to the Chinese gigacity in its characteristics. In addition, the western Europe (population 197 M) and the BosWash megalopolis area (population ~50 M) in the north-east US are both still predicted to have an increasing population density in the future (Hoornweg and Pope, 2017). These areas could show a degree of connectivity between the cities, where the problems cannot be solved by one municipality or even one country acting alone. In the future also the north shore of the Gulf of Guinea will be another potential gigacity area especially around Lagos, which is projected to be the largest city in the world (~100 million people) by the end of the century (Hoornweg and Pope, 2017).

The experiences gained from the Chinese gigacity could reveal a tipping point in which the whole area starts to act as one enormous urban area with interlinked problems instead of individual megacities. In addition, the critical areas of development and the number of observation stations needed to understand the changes induced could be identified and therefore tackle the upcoming problems. This increased knowledge gained from the Chinese gigacity can greatly benefit the other potential gigacity areas in the sustainable development of these regions in the future.

**6 Required actions and the roadmap**

In order to have an integrated research program covering all the scales, we first need to expand the network of comprehensive research stations and provide the data freely available to whomever might need it. We need enough measurement stations to capture the spatial variation of different phenomena and circulation of air within the individual megacities as well as within the whole gigacity and the surrounding regions.

To establish an integrated comprehensive research program is the key to go beyond the present ongoing research by collecting and utilizing openly available big data from in situ, satellite and models. Instead of research focusing on individual sites and phenomena, we will perform interlinked investigations including all the interactions between different scales, feedbacks, meteorology, hydrology, air pollution etc.

A framework is needed, in which a multidisciplinary scientific approach has the required critical mass and is strongly connected to dynamic policy making. Dynamic proactive policy making requires a lot of reliable data through a chain of harmonized network of observations. This is true in a case in fighting against global grand challenges, including climate change, air quality and viruses like COVID-19. The following items are proper actions to proceed according to the roadmap:

**i) From unknown to data**. It is crucial to collect comprehensive data. These include environmental variables, health and socioeconomic data as well as behavioral patterns and movement of people. We need to continuously harvest data (Kulmala, 2018), even for the research questions we are not yet aware but may/will emerge in the future. With big, open data we can in future answer such questions that do not exist yet, but it will be possible only if the needed data will exist at that moment already.

    **ii) Platform and analytics.** We need a platform for integrative analytics. We have to focus on reliable ways to collect, share, analyze and integrate data. We have to apply artificial intelligence and take full advantage of modern telecommunication systems, such as 5G, to make rapid data synthesis beyond the current state of the art.

**iii) Innovative cooperation.** We need innovative and open cooperation on local, regional, national and international levels. An open discussion and different views are needed to understand our own aspects better. Globally, we need to strengthen multilateral cooperation. While a bigger player may have more influence, a smaller may have more resilience.

    Finally, the decision makers should take the responsibility to steer the process towards a sustainable gigacity with the steps

identified above. A comprehensive long-term research and innovation program supported by comprehensive data platform in environmental, societal, health, economic and policy data would support the sustainable urban development in the gigacity level and help the society to respond to the urgent challenges (e.g. the COVID-19 pandemic). Therefore, it is crucial to promote

this kind of program and start it as soon as reasonable. In the case of success, the gigacity has the potential to be a visible example of a sustainable future society.

## Acknowledgements

The work is supported by Academy of Finland via Center of Excellence in Atmospheric Sciences (project no. 272041) and European Research Council via ATM-GTP 266 (742206). This research has also received funding from Academy of Finland
(project no. 316114 & 315203, 307537), Business Finland via Megasense-project, European Commission via SMart URBan Solutions for air quality, disasters and city growth, (689443), ERA-NET-Cofund as well Jane and Aatos Erkko Foundation and Academy of Finland Flagship funding (grant no. 337549). Partial support from the National Key R&D Program of China (2016YFC0200500), and the National Natural Science Foundation of China (91544231 & 41725020) is acknowledged. Leena Järvi and Pauli Paasonen are acknowledged.

## Author contribution

MK and AD conceived the paper. All of the authors contributed to the writing of the manuscript.

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

**Scope of Global Grand Challenges**

Global warming

Earthquakes

Climate change

Air quality

Fresh water

Volcanoes

Ocean acidification

Energy

Deforestification

Epidemic diseases

Biodiversity loss

Chemicalisation

Food supplies

**Demography / Population / Urbanization**

Figure 1: The burning planet and the scope of interlinked grand challenges. The main driving forces behind these challenges are the growth of population and gross domestic production globally, as well as the growth of urbanization closely related to the two former trends. The growing population needs clean air, more fresh water, food and energy, which will cause challenges such as climate change, declining air quality, ocean acidification, loss of biodiversity and shortages of fresh water, food and energy supplies as well as regional and even global epidemic diseases.

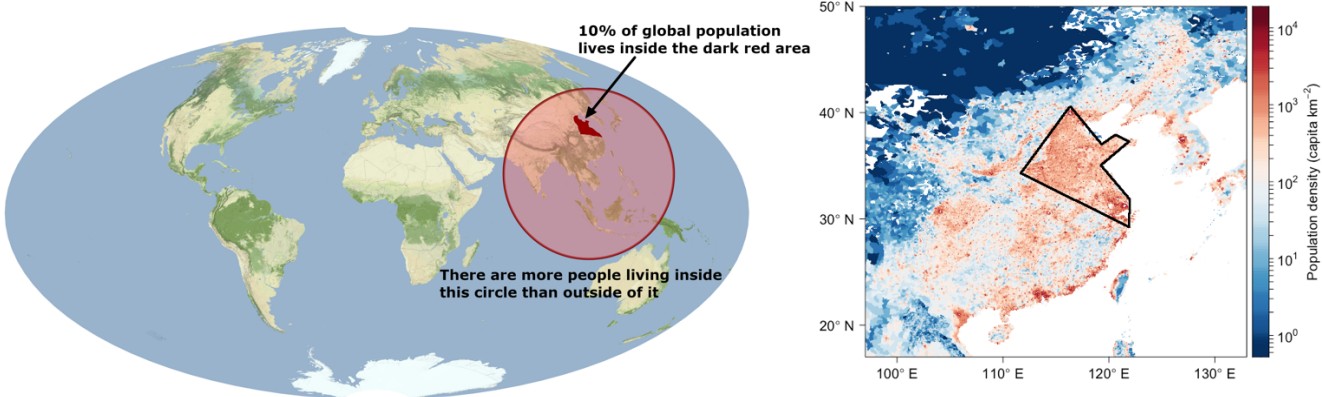

**Figure 2: Population density map including the Gigacity in East China. Half of the population of the World lives inside a circle that covers China, India and the South East Asian peninsula. Inside the triangle, there lives approximately 10 % of the World's population. This region is the most important economic motor of the world's economy and the GDP particularly in this Gigacity has increased very rapidly during the last decades. The data are obtained from Stamen Design (CC BY 3.0) and Gridded Population of the World (GPWv4.11; CC BY 4.0) (CIESIN, 2018).**

430

435

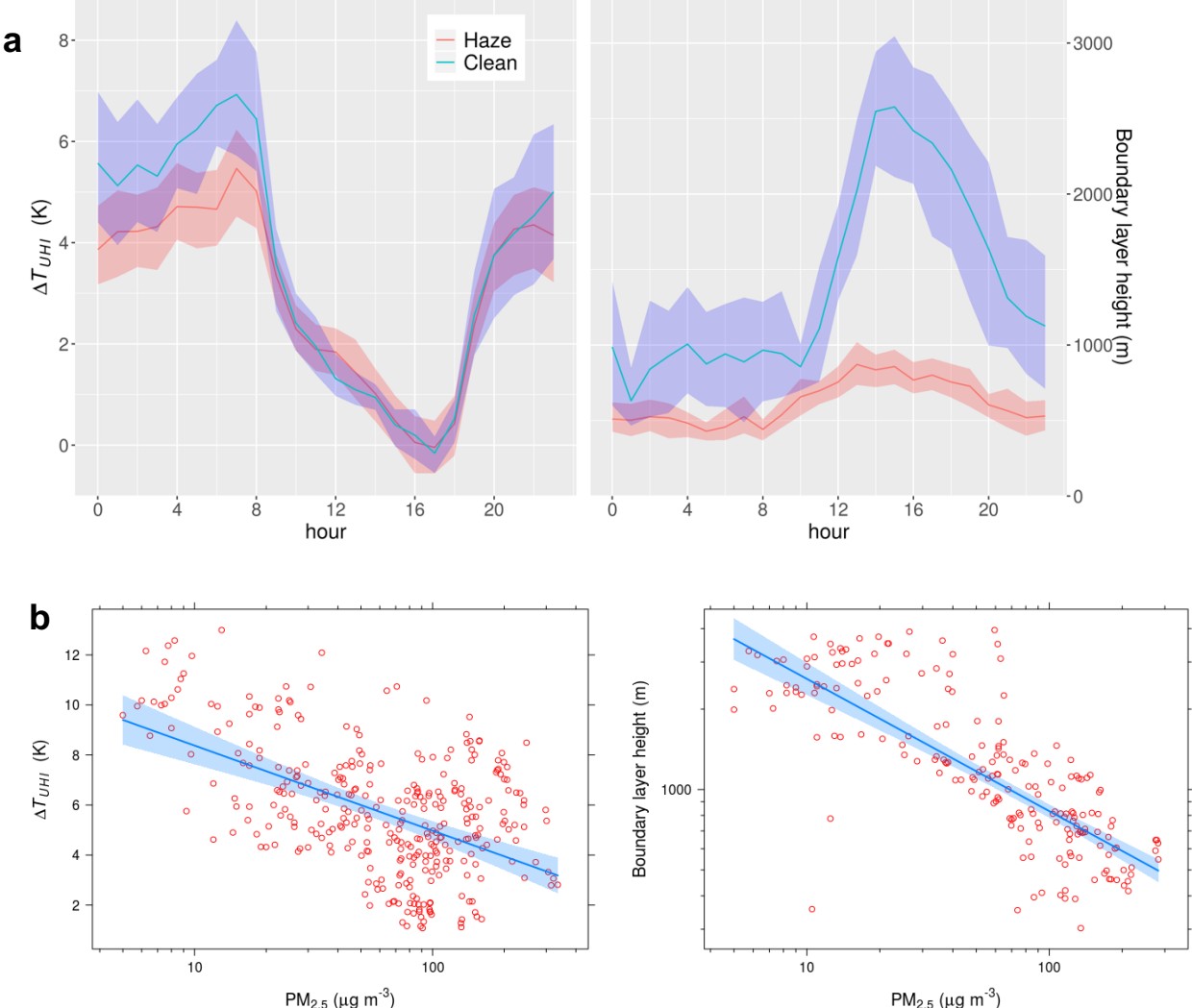

Figure 3: (a) Mean diurnal cycle of the magnitude of urban heat island ($\Delta T_{UHI}$) and boundary layer (BL) height during haze (30 days) and clean conditions (24 days) between 6 February 2018 and 31 March 2018. The shaded areas are the 95% confidence intervals of the fitted curve. $\Delta T_{UHI}$ is calculated as a difference of air temperature between the Beijing University of Chemical Technology (BUCT) and rural station at south-east of Beijing (Daxing, 39.654 °N, 116.695 °E, see Ma, et al., 2018). (b) Scatter plots of boundary layer height versus PM$_{2.5}$ concentration for daytime (hours 14–18) (right panel) and urban heat island magnitude ($\Delta T_{UHI}$) versus PM$_{2.5}$ concentration for nighttime (hours 0–8) (left panel). The values where $\Delta T_{UHI}$ is under 4 K with low PM$_{2.5}$ concentrations (<40) has been filtered out since those low values of $\Delta T_{UHI}$ are presumably caused by other phenomena than haze (e.g. cloudiness). The shaded areas indicate 95% confidence intervals.

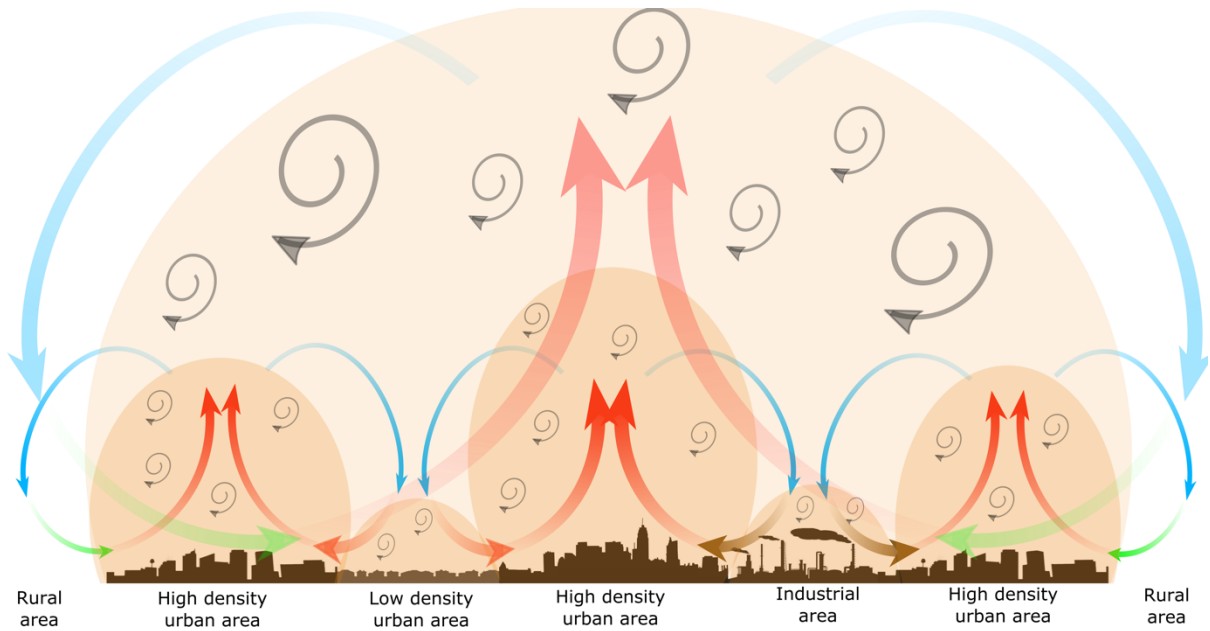

| Rural area | High density urban area | Low density urban area | High density urban area | Industrial area | High density urban area | Rural area |

**Figure 4: A schematic figure showing a complex structure of Urban Heat Island (UHI) in the gigacity, where within one large UHI, multiple individual UHI circulation patterns are formed between the lower and higher-density urban areas with the lack of rural areas. Within the gigacity, the UHI circulation is not bringing cooler and cleaner air from rural areas but draws in already warmer air from the lower-density urban areas and even highly-polluted air from the industrial areas. Only at the sides of the gigacity, cooler and cleaner air is drawn into the city and it will not penetrate into the middle parts of the gigacity. If the temperature difference between the high-density urban areas in the middle and the lower-density urban areas is not sufficient, the UHI circulation might also be totally suppressed in the middle parts of the gigacity. The complex internal circulation can lead to high ambient temperatures and high concentration of atmospheric pollutant levels inside the gigacity.**

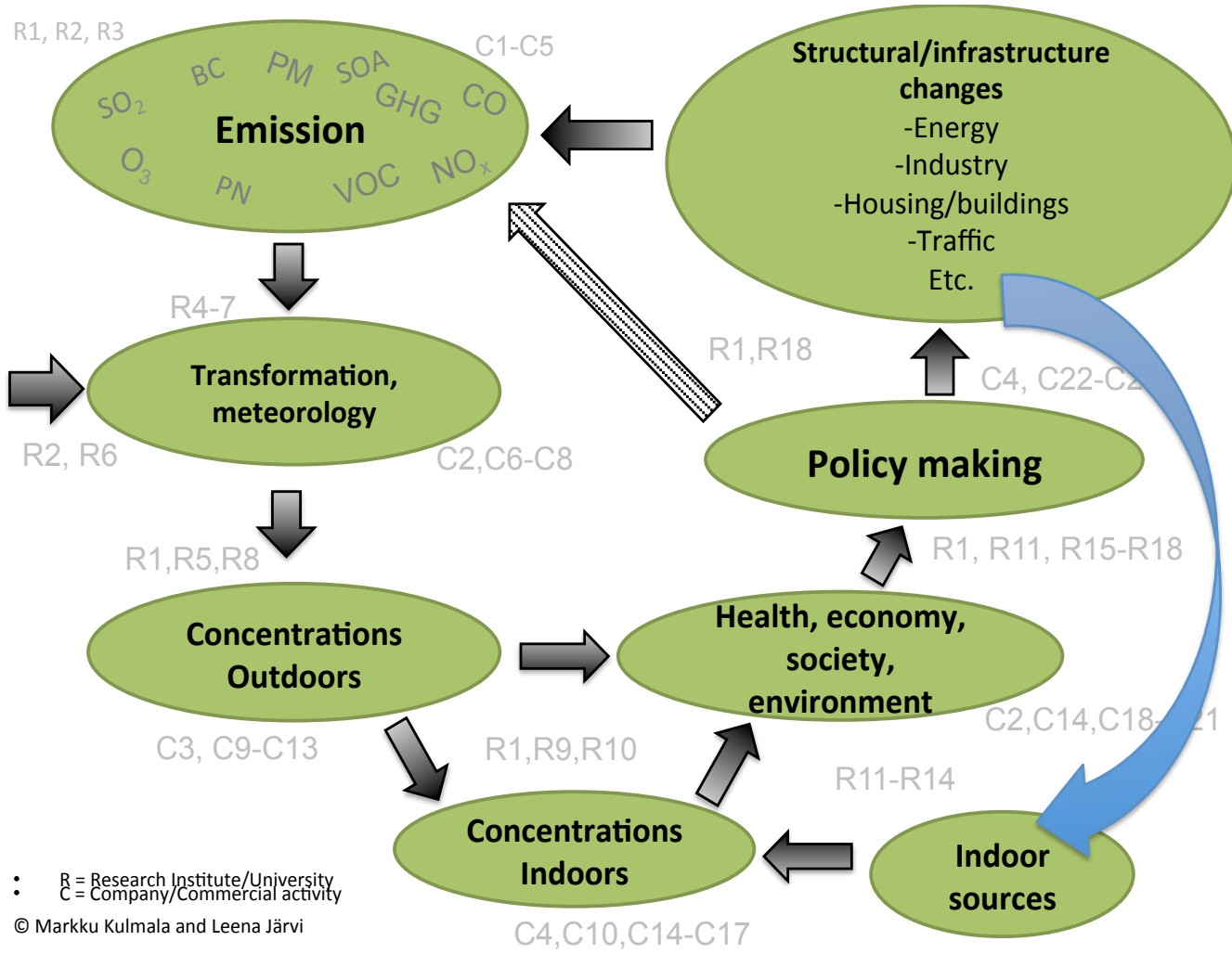

460

**Figure 5: Schematic figure for a holistic research and innovation program. Emissions include all potential emissions. The arrow from the left to the "transformation, meteorology" box means different air masses. The arrows from concentrations (indoors/outdoors) lead to the impact box, and therefrom to decision making. There could be direct decisions to cut emissions or indirect ones with changing structure/infrastructure.**

465