# Peer review of "Opinion: Gigacity – a source of problems or the new way to sustainable development"

_Atmospheric Chemistry and Physics, 2020_

## Author Comment (AC1)

We want to thank the reviewers for their comments on our manuscript "Opinion: Gigacity – a source of problems or the new way to sustainable development". These changes have improved the paper. Our detailed responses are given in blue below. Page and line numbers in our responses are referring to the "track changes" -version of the revised manuscript.

**Roy M. Harrison:**

Kulmala and co-authors provide a fascinating insight into the consequences of the huge concentration of population living in eastern China. The demographic statistics are remarkable: 10% of the world's population living in a 1Mkm$^2$ area of eastern China, and more than 50% of global population concentrated in an area of South-east Asia covering only a small proportion of the global landmass. The paper makes a strong case that the "gigacity" in eastern China greatly affects local weather patterns and may influence global climate through the Asian monsoon. Such influences appear inevitable, and a strong case is made for further research.

Thank You very much for Your kind statement.

1. In formulating such a research programme many factors need to be considered. While urban Beijing is clearly very densely populated, travelling only a few kilometres beyond the boundaries of the city reveals open countryside with evidently low population density. The extent to which this rural hinterland moderates the impact of the built-up areas upon the weather and climate is a key question, as is the consequence of further development of this rural space, which will surely fill gradually if population expands further.

It is true that especially north of Beijing is a low-population rural area, which is also outside the gigacity area. Even inside the gigacity there are still less urbanized, or even rural, areas between the individual cities. However, as you also pointed out, these areas are gradually filling and therefore the consequences should be carefully studied to have a better understanding of the problems that it may cause in the future. In the paper we also mentioned this, and the sentence was rephrased to emphasize it further (P2 L44-46): "The population is still increasing and gradually filling the less urbanized areas between the individual cities, so that this region is becoming practically one city – termed gigacity here – particularly from the grand challenges point of view".

2. The paper shows the impact of haze pollution on mixing heights and explains how this serves to exacerbate ground-level pollution by reducing vertical mixing. It appears to imply that the haze is an inevitable consequence of the gigacity, a point on which my opinion would differ. An analysis of urban/rural gradients in pollutant concentrations in Beijing (Harrison et al., 2021) demonstrates that for many pollutants, the high concentrations are a regional phenomenon and little influenced by emissions within Beijing itself. However, this is not the case for all pollutants, and most notably for nitrogen dioxide which shows marked urban and roadside increments above the rural concentrations. This demonstrates that local measures can improve air quality for some pollutants. Additionally, a careful analysis of air quality trends in Beijing which corrects for the influences of changing weather (Vu et al., 2019) shows the success of the 2013-2017 five-year pollution control plan in reducing all measured pollutants except ozone. This was achieved largely by enhanced controls on road traffic and domestic combustion, but much can still be done to reduce emissions from rural

biomass burning (Wu et al., 2020) and from industry, thereby reducing concentrations of long-lived primary pollutants such as carbon monoxide, and secondary particles formed from emissions of $SO_2$, $NO_x$ and VOC. Such measures, by reducing the haze phenomenon, will increase mixing depths and have huge benefits for air quality. The haze is not unique to China (consider Delhi which has a worse haze problem than Beijing), and nor is it an inevitable consequence of the gigacity, although such extensive urbanisation and industrialisation makes the control challenge inevitably greater.

We added text describing the complex mixture of local and regional pollution sources and included the Clean Air Act at the end of Section 3 (P4-5 L 115-130):

"The atmospheric pollution at the gigacity area is a complex mixture of local and regional-scale phenomena (e.g. Harrison et al., 2021; Kulmala, 2015; Wu et al., 2020). On one hand we have emissions from the local emission hotspots within the gigacity (e.g. industrial sources) as well as local and regional-scale secondary aerosol formation. On the other hand, we have regional transport of pollutants within the gigacity as well as transport from outside the gigacity area (e.g. industry, biomass burning etc.). The Chinese government has started to tackle the air pollution with the Clean Air Act from 2013 onwards, which has already shown substantial reduction of pollutants in eastern China (Vu et al., 2019). However, non-balanced reductions in otherwise polluted regions could lead even to a negative effect as was demonstrated during the COVID-19 lockdown (Huang et al, 2020a). Therefore, further research is needed in order to better understand the synergic effects of multi-pollutant emission reduction.

Even though the reduction of pollutants will lead also to increased BLH, which will reduce the pollution further, the cleaner air will allow also more incoming solar radiation to reach urban surfaces, which will increase the UHI effect in the gigacity area. Already now the individual megacities within the gigacity are suffering from extreme heat during the summertime (e.g. Nanjing). An increase of UHI and the lack of cooler air as an input flux from the diminishing rural areas could lead to unbearable urban summertime temperatures inside the gigacity. Therefore, a comprehensive research program would be needed in order to understand the interactions of the interlinked local and regional-scale phenomena, the associated feedbacks, and the highly complex atmospheric chemical processes."

3. Another point to consider is the point at which population size and density creates an entity which has properties similar to those of Eastern China. Although the population is far lower, the north-eastern United States (referred to as the BosWash megalopolis, with a population of over 50M), and western Europe (197M population) show a degree of connectivity in which air pollution phenomena cannot be solved by one municipality, or in Europe, one country acting alone. There is inevitably a continuum of urbanised land masses ranging from individual major megacities to the Chinese gigacity, and research needs to study the impacts of population size and density, and geographic extent upon weather, climate and air quality in the wider context of impacts upon human health and ecosystems.

We have included a section describing the global areas potentially falling into the gigacity criteria now or later on in future (Page 6-7 Section 5):

"5 Other potential gigacity areas
At the moment there are no other truly gigacity areas in the world where the cluster of individual megacities could be classified as one huge continuous urban area. However, there are already other possible areas known that might struggle with similar problems in the future as the Chinese gigacity. Especially the Ganges basin with a population of about 400 million people is definitely already quite close to the Chinese gigacity in its characteristics. In addition, the western Europe (population 197 M) and the BosWash megalopolis area (population ~50 M) in the north-east US are both still predicted to have an increasing population density in the future (Hoornweg and Pope, 2017). These areas could show a degree of connectivity between the cities, where the problems cannot be solved by one municipality or even one country acting alone. In the future also the north shore of the Gulf of Guinea will be another potential gigacity area especially around Lagos, which is projected to be the largest city in the world (~100 million people) by the end of the century (Hoornweg and Pope, 2017).

The experiences gained from the Chinese gigacity could reveal a tipping point in which the whole area starts to act as one enormous urban area with interlinked problems instead of individual megacities. In addition, the critical areas of development and the number of observation stations needed to understand the changes induced could be identified and therefore tackle the upcoming problems. This increased knowledge gained from the Chinese gigacity can greatly benefit the other potential gigacity areas in the sustainable development of these regions in the future."

**References**

Harrison, R.M., Vu, T.V., Jafar, H., Shi, Z., 2021.  More mileage in reducing urban air pollution from road traffic, Environ. Intl., 149, 106329.

**Vu, T.V.,  Shi, Z.,  Cheng, J.,  Zhang, Q., He, K.,  Wang, S., Harrison, R.M., 2019.**  Assessing the impact of clean air action on air quality trends in Beijing using a machine learning technique, Atmos. Chem. Phys., **19**, 11303-11314.

Wu, X., Chen, C., Vu, T.V., Liu, D., Baldo, C., Shen, X., Zhang, Q., Cen, K., Zheng, M., He, K., Shi, Z., Harrison R.M., 2020.  Source apportionment of fine organic carbon (OC) using receptor modelling at a rural site of Beijing: Insight into seasonal and diurnal variation of source contributions, Environ. Pollut., 266, 115078.

**Anonymous referee #2:**

The authors proposed a novel concept of gigacity, explains its great importance on regional air quality and global climate, and propose a large-scale research and innovation program to investigate its impact and the underlying mechanism. This is an important and very timely initiative and the answer to this question could be crucial for future planning of urbanization towards sustainable development.

Thank You very much for Your kind statement.

4. Gigacity is a nice concept and the authors have taken the eastern part of China as an example. It would be helpful if the authors could also explicitly define what is a gigacity in analogy to a megacity. Then people can tell if there is any other region falls into this category, e.g., eastern US, Europe, Indo-Gangetic Plain?

We have included a section describing the global areas potentially falling into the gigacity criteria now or later on in future (Page 6-7 Section 5). In this section the definition of the gigacity concept is also further explained (P6-7 L188-194): "At the moment there are no other truly gigacity areas in the world where the cluster of individual megacities could be classified as one huge continuous urban area. However, there are already other possible areas known that might struggle with similar problems in the future as the Chinese gigacity. Especially the Ganges basin with a population of about 400 million people is definitely already quite close to the Chinese gigacity in its characteristics. In addition, the western Europe (population 197 M) and the BosWash megalopolis area (population ~50 M) in the north-east US are both still predicted to have an increasing population density in the future (Hoornweg and Pope, 2017). These areas could show a degree of connectivity between the cities, where the problems cannot be solved by one municipality or even one country acting alone."

5. The authors have emphasized the importance of high-resolution modeling (10 meters) in forecasting air quality and suggests expanding this into a gigacity scale (100 km to more than 1000 km). But is 10 meters really necessary? According to the citations of Huang et al. (20202a and 2020b), where spatial resolution of a few tens of km has been adopted, I guess the authors mean 10 km here? If yes, I don't think that the resolution is really the limiting factor here in order to study the interactions and links in a gigacity scale. But on the other hand, to study the impact of gigacity within itself and on global climate, a direct application of the Earth System modeling would be the future. There how to increase the spatial resolution of Earth System models and reduce its dependence on sub-grid parameterization can be very challenging.

In order to tackle all the grand challenges, we need to take into account processes ranging from small-scale phenomena (e.g. local pollutant hotspots, local and regional scale secondary aerosols, local scale UHI etc.) to even global-scale phenomena (e.g. from Asian monsoon to global climate) and interactions between them. The global problems cannot be solved only by focusing on large-scale phenomena, since even those are interlinked to local-scale phenomena occurring inside and around the gigacity. Therefore, we would need a research program that takes into account all the scales and the results from local-scale studies should be coupled with larger-scale models that have a coarser resolution and not good enough description of urban areas and local phenomena.

In order to clarify this message, we have rephrased the sentence (P3 L75-77): "Although high-resolution modeling is capable of forecasting air quality and study the urban climate phenomena inside a megacity with a grid scale of about 10 meters, incorporating local-scale phenomena into a gigacity scale is not straight forward as it requires scaling from 100 km up to more than 1000 km (Huang et al., 2020a; Huang et al., 2020b)."

6. For the challenges specific for the gigacity, the authors may want to point out what questions have already been solved and what are the remaining open questions. For

example, the current regional and global models have already the capacity to simulate the Urban Heat Island effects in the gigacity scale. Maybe not perfect, but what would be processes and mechanisms that need to be improved or added there for the applications in a gigacity scale? Similar questions also apply to the aerosol-radiation-cloud interactions and other links.

The main concern is that we should be able to combine high-resolution modelling with larger-scale modelling and to take into account the associated feedbacks and interlinked phenomena (e.g. reduction of pollution leads to increased UHI). For example, using high-resolution modelling, we can examine the interaction of e.g. cooling pathways (like "green fingers") inside cities and the UHI circulation between the city and more rural areas surrounding it. Problems arise when we want to combine these fine scale phenomena for gigacity scale phenomena, where these fine-scale interactions and possible shrinkage of rural areas inside the gigacity affect the clean and cool air input into the cities. With modern high computing facilities these could be presumably combined, but this type of research, including both fine-scale and gigacity-scale phenomena, has not been made and therefore we try to emphasize the importance of a research program covering all the scales. Many of the Grand Challenges are falling into different scales, but they should be solved together since they are highly interlinked.

Some discussion related to this has been added to (P4-5 L124-130):

"Even though the reduction of pollutants will lead also to increased BLH, which will reduce the pollution further, the cleaner air will allow also more incoming solar radiation to reach urban surfaces, which will increase the UHI effect in the gigacity area. Already now the individual megacities within the gigacity are suffering from extreme heat during the summertime (e.g. Nanjing). An increase of UHI and the lack of cooler air as an input flux from the diminishing rural areas could lead to unbearable urban summertime temperatures inside the gigacity. Therefore, a comprehensive research program would be needed in order to understand the interactions of the interlinked local and regional-scale phenomena, the associated feedbacks, and the highly complex atmospheric chemical processes."

and (P7 L209-212):

"To establish an integrated comprehensive research program is the key to go beyond the present ongoing research by collecting and utilizing openly available big data from in situ, satellite and models. Instead of research focusing on individual sites and phenomena, we will perform interlinked investigations including all the interactions between different scales, feedbacks, meteorology, hydrology, air pollution etc."

7. For the proposed research program, the authors may want to further explain how it would differ from previous/current research programs on air quality studies in megacity or at a regional scale. For example, for the observation network, does it only differ by the number/density of stations?

The largest difference would be the comprehensive research stations with openly available data with high enough spatial coverage within and around the gigacity and taking into account all the scales (from local to even global) as pointed out in many parts of the text (e.g. P5 L133-135; P5 L155-157; P7 L204-207; P7-8 L219-223).

This was also further clarified (P7 L209-212): "To establish an integrated comprehensive research program is the key to go beyond the present ongoing research by collecting and utilizing openly available big data from in situ, satellite and models. Instead of research focusing on individual sites and phenomena, we will perform interlinked investigations including all the interactions between different scales, feedbacks, meteorology, hydrology, air pollution etc."

8. "Since this gigacity has roughly 50 times more people and 60 times larger surface area than Beijing – a typical megacity – its future is crucial not only for local people but also globally. The area is a huge emitter of greenhouse gases and air pollution as well as a potential source for local, regional and global epidemiological diseases." Due to the high population density, the region or rather the cities there might become potential hotspot, but not necessarily the source for local, regional and global epidemiological diseases. In view of the few cases and efficient control of COVID-19 in the region, as well as the significant improvement of the air quality there in recent years, I would like to second and add on to the statement from the other reviewer that the haze and/or the spread of an epidemic are not an inevitable consequence of the gigacity, although such extensive urbanisation and industrialisation makes the control challenge inevitably greater.

We have added text where we describe briefly that some reductions of emission have already been made by the Chinese government. However, we also pointed out that the enhanced secondary aerosol formation during the COVID restrictions shows that reducing the air pollution is a complex phenomenon that needs further research (see answer to comment #2).

There are indications that the loss of biodiversity favors the emergence of new infectious diseases and therefore any gigacity area in the world could be a potential source for new epidemiological disease as also pointed out in the text (P2 L38-39). The sentence you pointed out was rephrased to emphasize this (P2 L50-52): "The area is a huge emitter of greenhouse gases and air pollution as well as a potential source for local, regional and global epidemiological diseases due to loss of biodiversity and a huge population density."

Minor:

9. "with increasing risks of cardiovascular diseases and diabetes, violence and injuries, outbreaks of infectious disease, like the current pandemic of COVID-19 (Huang et al., 2020a; Tian et al., 2020), and inequity between people living in urban areas." The reference Huang et al., 2020a is not supporting the statement about increasing risks of diseases.

The reference for Huang et al. was taken off from here.

10. "It is already well known that reduced concentrations of nitrogen oxides will, under otherwise polluted conditions, increase ozone production and increase secondary aerosol particle concentrations (Ding et al., 2013, Liu and Tang, 2020)." Many studies show that reducing NOx will lead to a reduction of PM.

Even though it can improve the air quality in some extent, the non-balanced reduction of some pollutants in otherwise polluted environment might not be enough as found out during the COVID lockdown, when Beijing still suffered from haze even with significant reductions of pollutants. This has been also pointed out in the text (P3 L65-68).

11. "If suitable observations are lacking, the option is to run the models using reanalysis data." Observations are also needed to produce the reanalysis data. The authors may want to explain a bit more on this.

This sentence was rephrased as (P5 L151-152): "If suitable observations are lacking high enough spatial coverage or some of the needed variables, the option is to run the models using reanalysis data."

12. "We need to continuously harvest data (Kulmala, 2018), even if we do not yet know for sure that we need it." This sentence should be reformulated. It sounds like we still need to do some measurements without a motivation, but I guess the authors does not mean that.

We actually do mean here that we should make measurements also for the research questions that we do not know yet. This sentence was rephrased in order to clarify this (P7-8 L220-223): "We need to continuously harvest data (Kulmala, 2018), even for the research questions we are not yet aware but may/will emerge in the future. With big, open data we can in future answer such questions that do not exist yet, but it will be possible only if the needed data will exist at that moment already."

---

## Author Response (AR2)

We are pleased to hear that our responses to the comments of the two reviewers have been satisfying. We would also like to thank the editor for the additional comment that helped us clarify the specific new challenges and the methods needed for the gigacity area. Our detailed responses are given in blue below. The page and line numbers are referring to the "track changes" -version of the revised manuscript.

Comments from the editor:
Your Opinion has been well received by two reviewers and I am satisfied with your responses, which have clarified some details and expanded the discussion to include some wider considerations.

We would like to thank the editor and the reviewers for their kind words.

I would like to add one more point for you to take account of before acceptance. You make a strong point that modelling needs to consider all scales. What is not clear is why this is particularly the case for a Megacity environment. The handling of sub-grid-scale phenomena has been at the heart of large-scale model development since the first models were developed. Almost all key processes in large-scale models are described using sub-grid parameterizations, such as clouds, radiation and turbulence. I find it a somewhat 'lazy' assertion that we need to consider sub-grid-scale processes in relation to Megacities. I also find that your Opinion does not adequately address how that should be done and what the specific challenges are. Ultimately it will require parameterizations, which has been the focus of probably 90% of model development for several decades. So the onus is on you in the Opinion to spell out more clearly what the specific new challenges are.

Recent studies have shown that the impact of anthropogenic processes, such as those related to air pollution, are not well described in the larger scale models by the current parameterizations. Therefore, there are notable biases e.g. in the prediction of free-tropospheric air temperature in the gigacity region. In addition, there are most likely interaction between the different megacity clusters within the gigacity, and therefore we need to study the interactions of the local-scale phenomena with the larger scale circulation and boundary layer dynamics. We added some more explanation to clarify this in the text (P4 L118-125):

"Many recent studies (e.g. Huang et al., 2018) have demonstrated that the impact of anthropogenic processes, such as those related to air pollution, have not been included in the nowadays weather forecast models. As a result, these models show notable biases in the prediction of free-tropospheric air temperature in the gigacity region. In fact, in the gigacity region, anthropogenic aerosols (e.g. black carbon) could significantly influence the development of PBL via not only the reduction of surface solar radiation but also its "dome effect" by heating the upper-PBL (Ding et al., 2016; Wang et al., 2018). In addition, aerosol-PBL feedback could also occur at the gigacity scale by amplifying the transboundary transport of haze, including secondary pollutants, between different megacity clusters, such as the Yangtze River Delta and the Beijing-Tianjin-Hebei Area (Huang et al., 2020)."

In addition, some of the processes might be unique to the gigacity region and therefore are not well described in the current models and their parameterizations. Therefore, we need

comprehensive observations with high enough spatial coverage and concurrent fully-coupled models in all scales, so that we could characterize the complex processes and their interactions in different scales. We have now clarified this in the revised text (P6 L157-160):

"Currently, detailed studies within the gigacity are typically covering the few different phenomena, such as weather forecast, urbanization or air quality, as separate issues. As mentioned above, in gigacity areas human activities could influence the climate system all the way from the nanoscale to the global scale."

And (P6 L169-179):

"Therefore, we need comprehensive observations and high-resolution fully-coupled models to accurately describe the molecular-scale chemistry, microphysics of aerosol-cloud-interaction, and the multi-scale processes of aerosol-PBL-weather feedback at the gigacity and even global scale. The spatial coverage of observations and the model resolution needs to be high enough to well characterize the complexity of emissions and chemistry, land-surface processes, and their interaction with the PBL meteorology. We also need to understand aerosol physics and atmospheric chemistry in molecular and nm scales, as the majority of particle number and mass concentrations, as well as ozone, is caused by secondary processes taking place in the atmosphere (e.g. Kulmala et al., 2021). To understand such secondary processes, we need proper observations and process level models – including quantum chemistry (e.g. Kulmala et al., 2021). Since some of the processes might be unique to the gigacity area, and possibly still unknown, we need comprehensive observations and concurrent models that cover different regions of the gigacity (Kulmala 2015, 2018), in order to characterize different processes and their interactions and to force the models and validate the results."

Also, some smaller changes have been made to make the text more compatible with the revised version (P5 L130, P5 L153-155, P6 L164-167)